# Stylistic Shifts in Human–LLM Conversations: Challenges and Adaptation

**Fulei Zhang**
Amazon.com Inc.
United States
zhanfule@amazon.com

**Zhou Yu**
Amazon.com Inc.
United States
amznzya@amazon.com

## Abstract

As Large Language Models (LLMs) are increasingly deployed in customer-facing applications, a critical yet underexplored question is how users communicate differently with AI-driven chatbots compared to human associates. In this study, we present empirical evidence that users adopt distinct communication styles when interacting with chatbots versus human representatives. Our analysis reveals significant differences in grammatical fluency, politeness, and lexical diversity between the two settings. These findings suggest that models trained exclusively on human-human interaction data may not adequately accommodate the communication style shift that occurs once an LLM chatbot is deployed. To enhance LLM robustness to post-launch communication style changes, we experimented with two strategies: (1) data augmentation during the post-training phase and (2) inference-time user message reformulation. Our results indicate that models trained on stylistically diverse datasets significantly outperform those trained exclusively on original or stylistically uniform datasets, while inference-time reformulation proved less effective. These insights help us to better adapt our models for improved LLM-user interaction experiences.

## 1 Introduction

Large Language Models (LLMs) offer considerable promise for task-oriented dialogue systems, demonstrating strong capabilities in intent understanding, context retention, commonsense reasoning, and the generation of human-like responses that enhance user experience. In industry conversational chatbot applications, LLM-powered assistants are typically developed and evaluated using historical human-to-human chat transcripts. However, one foundational question often goes unexamined: Do users communicate in the same way with LLM virtual assistant as they do with human associates?

According to Communication Accommodation Theory (CAT), people naturally adjust their communication style to match or mirror their conversation partners during interactions[Dragojevic et al., 2015]. While LLMs are capable of producing fluent responses, their perceived non-human identity and stylistic tendencies may prompt users to adopt a different linguistic style. As a result, user messages in human-LLM interactions may diverge from those in human-human settings - potentially affecting system performance, particularly when models are trained predominantly on human-human data.

This study addresses this gap by analyzing user messages during intent understanding in both human–human and human–LLM conversations. Our contributions are: (i) quantifying linguistic variation across six stylistic and semantic dimensions; (ii) proposing style-aware data augmentation with minimal and enriched rewrites; (iii) demonstrating that incorporating stylistically diverse training data significantly improves performance on human–LLM inputs; and (iv) finding that inference-time

39th Conference on Neural Information Processing Systems (NeurIPS 2025) Workshop: Reliable ML from Unreliable Data.

style normalization is less effective. To our knowledge, this is the first empirical study of linguistic adaptation to conversational AI, providing practical strategies for more robust and adaptive LLMs.

## 2 Related Work

Despite the growing popularity of Large Language Models (LLMs) and their promising potential in task-oriented chatbot services such as travel planning, customer support and sales, there remains a significant gap in understanding how users modify their communication patterns when interacting with AI assistants versus humans. Prior work shows that user style shifts when human involvement is disclosed in hybrid chat services [Gnewuch et al., 2024], with messages becoming longer, more complex, and denser. However, these findings may not fully capture dynamics in purely LLM-driven settings. Research on LLM robustness to noisy inputs such as ASR errors, typos, and irregular syntax reports mixed outcomes, ranging from strong resilience [Singh et al., 2024] to clear degradation [Wang et al., 2024] - depending on noise type and task. To address input variability, post-training augmentation techniques [Meng et al., 2021] teach models to recognize semantic equivalence, while inference-time methods attempt to clean or correct inputs [Zhang et al., 2024, Jiang et al., 2024]. Our work differs by focusing on naturally occurring stylistic divergence between human–human and human–LLM interactions as a form of distribution shift.

## 3 Experiments and Results

**Setup.** We studied the task of intent understanding during e-commerce task oriented dialogues: given a user's first message in a dialogue, predict its intent from a fixed ontology. Our data includes 13K human–human conversations, used for model training and generating stylistically diverse variants, and 3.7K human–LLM conversations, used solely to analyze stylistic differences. For evaluation, we used a separate set of 1.3K human–LLM utterances with annotated intent labels. This test setting reflects a realistic deployment context where models trained on human-human dialogue are applied to human-AI interactions. Non-informative greetings were excluded from the dataset. At inference time, the model receives a standardized instruction, a list of candidate intents, and the user message, and is expected to generate the label that best reflects the user's need. Accuracy is measured as exact match with the annotated intent.

**Human–LLM vs. Human–Human Interaction Language Divergence.** We evaluated user utterances from both the human–human and human–LLM conversation datasets across six linguistic dimensions: grammar fluency, politeness/formality, lexical diversity, informativeness, clarity, and emotion intensity. Each dimension is formally defined below. Each user message was evaluated on a 1–5 scale using the Claude 3.5 Sonnet v2 model, guided by chain-of-thought prompting to ensure consistent and calibrated judgments across utterances. The complete rubric and scoring prompt is provided in Appendix A.1.

- **Grammar Fluency:** Are the grammar and sentence structure fluent and correct?
- **Politeness/Formality:** Is the tone polite or formal (e.g., "please," "thank you")?
- **Lexical Diversity:** Does the user use varied and rich vocabulary?
- **Informativeness:** Does the message provide actionable or detailed information relevant to resolving the issue?
- **Explicitness/Clarity:** Is the request clearly stated or vague/ambiguous?
- **Emotion Intensity:** How strongly is the user's emotion expressed (e.g., frustration, urgency)?

As shown in Table 1, user messages to LLM assistants were significantly less fluent (-5.3%), less polite/formal (-14.5%), and slightly less diverse (-1.4%) than those to human associates. No significant differences were found in informativeness, clarity, or emotional intensity. These results suggest that users adjust their linguistic style based on the nature of the recipient - being more formal, polite, and grammatically complete with humans, creating a stylistic domain shift between human-human and human-AI conversations. [1]

---

[1]Due to data privacy constraints, we are unable to release the dataset or source code. All reported metrics are presented as relative changes (deltas) to preserve confidentiality.

Table 1: Comparison of linguistic dimensions between human–LLM and human–human interactions. Relative differences are reported as percentage changes (negative = lower in human–LLM). Stars denote significance levels (* $p < 0.05$, ** $p < 0.001$).

| Linguistic Feature | Relative Difference (%) | $p$-value |
|---|---|---|
| Grammar Fluency | -5.3 | $< 0.001^{**}$ |
| Politeness/Formality | -14.5 | $< 0.001^{**}$ |
| Lexical Diversity | -1.4 | $0.0378^{*}$ |
| Informativeness | +0.5 | 0.535 |
| Explicitness/Clarity | +0.5 | 0.399 |
| Emotion Intensity | +0.9 | 0.258 |

**Post-Training with Diverse Linguistic Style-augmented Data.** To address the linguistic style mismatch identified, we propose a post-training phase mitigation strategy that synthesizes new data reflecting diverse stylistic variants. We synthetically rewrote human–human utterances into Minimal (terse, ungrammatical) and Enriched (formal, fluent) styles using controlled prompting. Rewrites were generated by adjusting each message to align the original linguistic scores $s = (s_g$ for grammar, $s_p$ for politeness, and $s_l$ for lexical diversity) to the target scores $t$. The rewriting procedure is formalized in Algorithm 1. Full prompt templates for both the Minimal Style and Enriched Style rewrites are provided in Appendix A.2 and Appendix A.3, respectively. The following example highlights the lexical, grammatical, and tonal shifts produced by our rewriting prompts.

> **Original message ($s_g$=3, $s_p$=3, $s_l$=3)**
> Hi, I'm looking to plan a trip to Paris next month. Can you help me find good flight and hotel options?
>
> **Minimal style rewrite ($s_g$=1, $s_p$=1, $s_l$=1)**
> paris next month. flights hotels?
>
> **Enriched style rewrite ($s_g$=5, $s_p$=5, $s_l$=5)**
> Good afternoon! I'm planning a vacation to Paris in the coming month and would appreciate your help finding the best deals on both flights and accommodations. Thank you!

---

**Algorithm 1** Controlled Rewriting Strategy

---

**Input:** User message $u$; style scores $s = (s_g, s_p, s_l)$; rewrite mode $m \in \{\text{MINIMAL}, \text{ENRICHED}\}$
**Output:** Rewritten message $u'$

**if** $m = \text{MINIMAL}$ **then**
    $t \leftarrow (\max(1, s_g-1), \max(1, s_p-1), \max(1, s_l-1))$
**else if** $m = \text{ENRICHED}$ **then**
    $t \leftarrow (\min(5, s_g+1), \min(5, s_p+1), \min(5, s_l+1))$
**end if**
Prompt LLM with $(u, s, t)$ using the rewrite template
$u' \leftarrow$ LLM output
**return** $u'$

---

This yielded four datasets: $D_1$ (original), $D_2$ (Minimal), $D_3$ (Enriched), and $D_4$ (Combined, union of $D_1$, $D_2$, and $D_3$). We re-scored each dataset using our rubric scoring prompt in Appendix A.1. As shown in Table 2, this retrospective analysis confirms that rewrites reliably shifted the linguistic characteristics in the intended directions. We fine-tuned Mistral-7B [Jiang et al., 2024] with LoRA [Hu et al., 2022] on each of the four datasets and evaluated on 1.3K human–LLM messages.

As shown in Table 3, the model trained on the combined dataset ($D_4$) achieved the best performance, with a **+2.9%** relative improvement over the baseline ($D_1$). This suggests that exposure to a range of linguistic styles, including formal, informal, and terse utterances, enhances the model's ability to generalize to real-world chatbot inputs. By contrast, models trained solely on minimal-style ($D_2$) or enriched-style ($D_3$) data underperformed relative to the baseline, with **–2.6%** and **–1.8%** drops in accuracy, respectively. Despite $D_2$ stylistically resembling typical language in user-LLM interactions, its narrow style range reduced generalization. These findings highlight the importance of stylistic

diversity in training data: intent detection systems perform best when exposed to a broad spectrum of real-world user expression rather than a single linguistic register.

Table 2: Relative differences (%) in linguistic dimension scores across training dataset variants. Positive values indicate higher scores compared to the human–human baseline ($D_1$).

| Dataset | Grammar Fluency | Politeness/Formality | Lexical Diversity |
|---|---|---|---|
| $D_1$ (Human–human) | 0.0 | 0.0 | 0.0 |
| $D_2$ (Minimal style) | −15.8 | −18.7 | −12.8 |
| $D_3$ (Enriched style) | +56.9 | +67.5 | +47.7 |
| $D_4$ (Combined) | +35.3 | +44.5 | +31.1 |

Table 3: Change in intent detection accuracy on human–AI inputs, relative to the baseline model trained on human–human data ($D_1$). Positive values indicate accuracy gains. Stars denote significance levels (* $p < 0.05$, ** $p < 0.001$).

| Training Dataset | $\Delta$ vs. $D_1$ (%) |
|---|---|
| $D_1$: Human–human | 0.0 |
| $D_2$: Minimal style | −2.6 |
| $D_3$: Enriched style | −1.8 |
| $D_4$: Combined | **+2.9**[**] |

**Reformulate Human-LLM Assistant Query at Inference Time.** To mitigate the domain mismatch caused by stylistic variation, an alternative to training-time augmentation is to align user input style at inference time., while still rely on models trained only on human-human interactions. This approach aims to convert LLM interaction-style queries into human interaction-style variants prior to prediction.

To apply this idea, we used the model trained solely on human-human data ($D_1$) and applied a controlled rewriting process to test inputs from human-AI conversations. Each user message was firstly scored along grammar fluency, politeness/formality, and lexical diversity dimensions using our rubric-based evaluator. Messages already resembling human–human style (i.e., above the threshold across all dimensions) were kept unchanged. Otherwise, we sampled a target style score from the $D_1$ distribution and used Claude 3.5 Sonnet v2 to rewrite the message to match the target style while preserving the original meaning and intent.

As shown in Table 4, inference-time rewriting resulted in a **–1.9% drop** in performance compared to using the original inputs. This result suggests that simply restyling input text to match the training data distribution may fail to preserve subtle intent-relevant signals present in original user messages. In some cases, rewriting may introduce unnatural phrasing or obscure key cues critical for classification.

Table 4: Impact of inference-time query reformulation on intent detection accuracy. Values shown as percentage change relative to the original human–AI input, negative values indicate accuracy drops.

| Inference Input Style | Accuracy $\Delta$ (%) |
|---|---|
| Original human–AI input | 0.0 |
| Rewritten to human–human style | **–1.9** |

## 4    Conclusion and Future Work

Through in-depth analysis of task oriented conversations from both user-LLM and user-human interactions, we identified user messages exhibit significant communication style differences in these two settings. We quantified these differences across six dimensions, finding substantial variations in grammatical fluency, politeness/formality, and lexical diversity. To better adapt models initially trained and evaluated on human-human communication data to the observed shifts in user communication style post-deployment, we conducted experiments using an intent detection task, exploring both post-training data augmentation techniques and inference-time message reformulation approaches. Our results demonstrate that increasing stylistic diversity in post-training data significantly improves

model performance on user-LLM assistant conversations, while inference-time message reformulation proves less effective. This study provides valuable insights into accommodating users' varied linguistic behaviors when interacting with LLM-based systems, enabling more robust conversational AI that can deliver optimal user experience. While our work primarily focused on the initial phase of conversation and intent detection tasks, future research should investigate how conversational AI can maintain engaging interactions throughout extended dialogues.

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

# A  Prompt Templates

## A.1  Language Scoring Prompt

```
I will give you a single user utterance. Your task is to evaluate the language
    used by the user. Use a chain-of-thought approach to reason through your
    judgments and output a structured JSON dictionary of scores.

Each score should be on a scale from 1 to 5, where 1 = very low / poor, and 5 =
    very high / excellent. For emotion categories, list the most likely one(s).

Evaluate the following dimensions:

1. Linguistic Features
   - GrammarFluency: Are the grammar and sentence structure fluent and correct?
   - PolitenessFormality: Is the tone polite or formal (e.g., "please", "thank you
       ")? Or informal/slangy?
   - LexicalDiversity: Does the user use varied and rich vocabulary?

2. Semantic Features
   - Informativeness: Does the utterance provide actionable or detailed information
       ?
   - ExplicitnessClarity: Is the request clearly stated or vague?

3. Emotional Features
   - EmotionIntensity: How strongly is the emotion expressed?

Think step-by-step. First, examine the grammar, politeness, and vocabulary. Then
    evaluate informativeness and clarity. Finally, assess emotional tone and
    intensity.
```

```
Return a JSON object only.

Begin reasoning now for the following utterance:
{{rewritten_text}}
```

## A.2 Minimal Style Rewriting Prompt (D₂)

```
You are a user message rewriting assistant. Your task is to rewrite user messages
    according to three language attributes while preserving the original meaning
    and informativeness.

Each attribute is rated from 1 (very low/poor) to 5 (very high/excellent):
  1. GrammarFluency: Are the grammar and sentence structure fluent and correct?
  2. PolitenessFormality: Is the tone polite or formal (e.g., "please", "thank you
     ")? Or informal/slangy?
  3. LexicalDiversity: Does the user use varied and rich vocabulary?

If the rewrite action is REWRITE:
  - Rewrite the message to reflect the target scores, especially when scores are
      low (e.g., 1 or 2).
  - Lower GrammarFluency = broken, fragmented, ungrammatical sentence.
  - Lower PolitenessFormality = no "please", "thanks", or polite phrasing.
  - Lower LexicalDiversity = repetitive, simple, blunt words.
  - The rewrite should be short, direct, and minimal if target scores are low.
  - Do not add or infer anything not in the original message.

If the rewrite action is KEEP:
  - Return the original message unchanged.

Output only the rewritten message. Do not explain or include any prefix or
    reasoning.

Original Message: {{processed_turn_text}}
Original Scores: GrammarFluency: {{grammar_fluency}}, PolitenessFormality: {{
    politeness_formality}}, LexicalDiversity: {{lexical_diversity}}
Target Scores: GrammarFluency: {{target_grammar_fluency}}, PolitenessFormality: {{
    target_politeness_formality}}, LexicalDiversity: {{target_lexical_diversity}}
Rewrite Action: {{rewrite_action}}
```

## A.3 Enriched Style Rewriting Prompt (D₃)

```
You are a user message improvement assistant. Your task is to rewrite user
    messages to improve their language across three attributes, while keeping the
     original meaning and intent unchanged.

Each attribute is rated from 1 (very low/poor) to 5 (very high/excellent):
GrammarFluency: Use fluent, grammatically correct, and complete sentence
    structures.
PolitenessFormality: Use polite or formal tone (e.g., "please", "thank you", "
    could you"), where appropriate.
LexicalDiversity: Use varied, expressive, and natural vocabulary.

When target scores are high (4 or 5), your goal is to:
- Improve sentence structure to be clear and fluent.
- Add softeners and polite language.
- Use more varied and natural vocabulary while preserving the original meaning.

Do not change the user's intent, add extra information, or make the message longer
     than necessary.
```

```
Only return the rewritten message. Do not explain your reasoning or include
    commentary.

Original Message: {{processed_turn_text}}
Original Scores: GrammarFluency: {{grammar_fluency}}, PolitenessFormality: {{
    politeness_formality}}, LexicalDiversity: {{lexical_diversity}}
Target Scores: GrammarFluency: {{target_grammar_fluency}}, PolitenessFormality: {{
    target_politeness_formality}}, LexicalDiversity: {{target_lexical_diversity}}
Rewrite Action: REWRITE
```

