# OpenReview forum: "Stylistic Shifts in Human–LLM Conversations: Challenges and Adaptation"
_NeurIPS.cc/2025/Workshop/Reliable_ML — NeurIPS 2025 - Reliable ML Workshop_

### Official Review · Reviewer_2xak · 2025-09-12
**Interesting study on stylistic differences between human-human and human-LLM interactions, but limited by motivation**

**Rating:** 4
**Confidence:** 4

**Review:**

1. Summary: The authors hypothesize that humans interact differently with LLM chatbots than with other humans. They suggest that this difference should be understood and incorporated into customer-facing applications. They find variations in linguistic styles across six dimensions and provide methods to transform prompts between different types. They show that (1) more stylistic diversity in post-training helps model performance while (2) message transformation techniques hurt model performance.


2. Strengths:
(A) The authors consider a large dataset (13k human-human vs. 3.7k human-LLM conversations).
(B) They collected novel datasets of rewritten prompts whose language is drawn from different distributions (original, minimal, enriched, and combined).
(C) The work takes the question of style seriously in considering LLM-human interactions. It provides evidence that the ways in which these entities communicate with each other are fundamentally different and explores ways to transform between different linguistic variations. This is a novel contribution!
(D) There is a balanced evaluation of strategies (post-training augmentation vs. inference-time reformulation).

3. Weaknesses:
(A) " In industry conversational chatbot applications, LLM-powered assistants are typically developed and evaluated using historical human-to-human chat transcripts." -> Is this true? It could be the case, for example, that the LLMs are graded and subsequently finetuned with respect to evaluation rubrics that intend to follow the company's Specs. The LLMs are trained on the corpus of Internet text and positioned for conversation via instruction fine-tuning and RLHF. It would be useful for you to examine how transcripts are typically used in the training and evaluation pipelines.
(B) Some of your rubric criteria (ie. lexical diversity) seem difficult to measure. What metrics did you use?
(C) The outcome in Table 4 (that intent accuracy drops under rewritten regimes) seems to provide evidence that the rewriting techniques are undesirable. Can you address this?

4. Suggestions:
(A) It would be useful for you to clarify what the labels ("annotated intent labels") on your conversational datasets indicate. Where did you source that datasets that you considered? Who completed the labeling?
(B)  It is not clear why it is necessary to match the style between human-human and human-LLM interactions. Do humans prefer one to the other? It would be useful to collect human preference judgments on this. The motivation section will be critical in this work.

5. Ethics:
(A) No ethical concerns.

---

### Official Review · Reviewer_ziKM · 2025-09-16
**Interesting short paper, issues on the experimental procedure**

**Rating:** 6
**Confidence:** 3

**Review:**

### 1. Summary:

The short paper builds on the idea that users communicate differently when they communicate with another human versus when they communicate with a chatbot. This may lead the chatbot to have worse performance than expected. They try two different approaches to mitigate this issue: (a) data augmentation during the post-training phase and (b) inference-time user-message reformulation. The first approach seems to have positive results, while the second one does not seem to be effective.

### 2. Strengths:

S1. The paper refers to a known problem of transfer learning (i.e., what happens when I test an ML system with data from different distributions of the training data), but with a fresh view: the psychological changes that lead people to communicate differently.

S2. They use intent classification instead of free-text answers, which makes the problem a classification one and, therefore, easier to evaluate.

S3. The results of both approaches feel logical and anticipated.  Things go better when the LLM is trained on data with higher diversity. However, the drop in performance in the D2 and D3 datasets is to be expected since it does not necessarily move the training dataset towards the "right" direction (as they are both extremes). Especially the second approach relies on the LLM to keep the correct intent when rewriting the messages, which will also have errors; therefore, the drop in performance is anticipated.

### 3. Weakness/Limitations.

W1. There is no standard procedure followed (e.g., k-fold cross-validation or another resampling technique) to make the results more robust. This could strengthen their conclusions.

W2. D4 is the combined dataset (i.e., three times the size of D1, D2, and D3).

W3. Average differences are not enough to decide what shifts should be made to the training data.

### 4. Suggestions.

Following W2, it would be better to present the results on D4 normalized by size (D4/3). Also, it would be nice to show how different sampling techniques could lead to better or worse results (highlighting the impact of distribution shift). Add more details on the experimental procedure followed in the Appendices. Minor presentation issues (move tables at the start/end of the pages, use a box around the prompt, etc.).

---

### Official Review · Reviewer_kdc3 · 2025-09-17
**Interesting and Timely Problem, but Better Execution Needed**

**Rating:** 5
**Confidence:** 3

**Review:**

Summary:

The paper studies stylistic differences between human–human (HH) and human–AI (HAI) interactions in customer-facing roles, finding reduced grammatical fluency, politeness, and lexical diversity in HAI conversations. To address this distribution shift, the authors propose augmenting training data with synthetic “Minimal” and “Enriched” rewrites of HH utterances. Experiments on intent detection show that models trained on the combined dataset perform better than those trained on any single style, suggesting that stylistic diversity improves robustness.

Strengths:

1. The paper addresses a novel and timely problem. As LLMs are increasingly deployed in customer-facing roles, understanding stylistic shifts in human behavior is both practically and academically relevant. These shifts can introduce distribution mismatch and lead to performance drops if models are not trained with such inputs.
2. The methodology and experiments are clearly presented. The proposed augmentation strategies and evaluation methods are easy to follow.

Weaknesses/Limitations:

1. The combined dataset (original + enriched + minimal) shows the best performance, but it is unclear whether this is due to stylistic diversity or simply increased dataset size. Without a size-controlled baseline (e.g., more original-style examples equal in size to the combined dataset), it is hard to attribute the gains conclusively. Moreover, the Minimal/Enriched datasets are synthetic (LLM-generated) and may introduce artifacts that explain their weaker performance compared to original data.

2. This is a bit minor, but I am not particularly convinced about the choice of intent detection on the first message as the sole evaluation task. While it provides a clear classification setup, it may not fully capture the practical consequences of stylistic shifts. Alternative or complementary metrics, such as the number of messages-to-resolution or multi-turn task success, could provide stronger evidence of impact in real settings.